# Numerical and Experimental Analysis of Dual-Beam Laser Polishing Additive Manufacturing Ti6Al4V

**DOI:** 10.3390/mi14091765

**Published:** 2023-09-13

**Authors:** Junyong Zeng, Wei Zhang, Ting Guo, Yan Lou, Wenqi Wang, Zhenyu Zhao, Chao Wang

**Affiliations:** 1School of Sino-German Robitics, Shenzhen Institute of Information Technology, Shenzhen 518172, China; zjy1972412357@163.com (J.Z.); 2013100925@sziit.edu.cn (T.G.); 18312215870@163.com (W.W.); 2College of Mechanical and Control Engineering, Shenzhen University, Shenzhen 518061, China; louyan@szu.edu.cn; 3School of Mechanical Engineering, Xiangtan University, Xiangtan 411105, China; 202021002210@smail.xtu.edu.cn

**Keywords:** dual-beam laser polishing, numerical model, surface roughness, molten pool, capillary, thermocapillary

## Abstract

Laser polishing is an emerging efficient technique to remove surface asperity without polluting the environment. However, the insufficient understanding of the mechanism of laser polishing has limited its practical application in industry. In this study, a dual-beam laser polishing experiment was carried out to reduce the roughness of a primary Ti6Al4V sample, and the polishing mechanism was well studied using simulation analysis. The results showed that the surface roughness of the sample was efficiently reduced from an initial 10.96 μm to 1.421 μm using dual-beam laser processing. The simulation analysis regarding the evolution of material surface morphology and the flow behavior of the molten pool during laser the polishing process revealed that the capillary force attributed to surface tension was the main driving force for flattening the large curvature surface of the molten pool at the initial stage, whereas the thermocapillary force influenced from temperature gradient played the key role of eliminating the secondary roughness at the edge of the molten pool during the continuous wave laser polishing process. However, the effect of thermocapillary force can be ignored during the second processing stage in dual-beam laser polishing. The simulation result is well in agreement with the experimental result, indicating the accuracy of the mechanism for the dual-beam laser polishing process. In summary, this work reveals the effect of capillary force and thermocapillary force on molten pool flows during the dual-beam laser polishing processes. Moreover, it is also proved that the dual-beam laser polishing process can further reduce the surface roughness of a sample and obtain a smoother surface.

## 1. Introduction

Laser polishing is widely used in metal and non-metal fields as a new type of surface treatment technology. Traditional polishing techniques have many weaknesses such as low polishing efficiency, high cost, environmental pollution, and difficulty polishing complex surface structures [1]. However, laser polishing overcomes these disadvantages and is widely used in aerospace, biomedical, automotive, and petrochemical fields due to its high processing precision, non-contact, and green environmental protection, which can greatly promote the development of laser polishing [1,2,3,4]. Due to its speed, low cost, high flexibility, and high integration advantage, SLM technology has been used to process complex structures parts by printing layer-by-layer, which has attracted a lot of attention from the industry [5,6,7,8]. Despite these advantages, SLM Ti6Al4V parts still suffer from high surface roughness, balling, stair-step effects, and adverse tensile residual stresses, which could negatively influence the wear resistance and fatigue life of the components [8,9]. Improving surface quality, such as eliminating surface defects and reducing surface roughness, is of important industrial value [10].

At present, various post-processing methods have been proposed to improve the surface quality of additive manufacturing parts, as listed in Table 1. Shen et al. [11] proposed a two-step laser polishing method to improve the surface quality of additively manufactured titanium alloy samples by combining a picosecond laser and a continuous laser, which reduced the surface roughness of the sample from 6.62 μm to 0.55 μm. In addition to surface roughness improvements, laser polishing can also increase the microhardness and wear resistance of additively manufactured Ti alloy, as reported by Xu et al. [12]. Wang et al. [13] used a multi-jet polishing (MJP) method to improve the surface quality of 316 L stainless steel prepared using powder bed fusion, which decreased the surface roughness from 0.84 μm to 0.03 μm. They also found that there was no obvious influence of MJP on the surface microstructure, surface composition, and material phase. In addition, Srishti et al. [14] used electrochemical polishing to improve the surface quality of SLM Incol718 alloy and obtained a finished surface of 0.25 μm. Ke et al. [15] proposed a ballonet polishing method to improve the surface quality of binderless tungsten carbide, which reduced the surface roughness from 104 nm to 3.7 nm. However, the application of this technology in components with complex shapes and surfaces is limited. Beaucamp et al. [16] used the shape adaptive grinding process to process the surface of titanium alloy samples, which reduced the surface roughness of titanium alloy samples to less than 10 nm.

In terms of theoretical study, many scholars have carried out experimental research on the flow of molten pools and the evolution of surface morphology using numerical simulation. Ma et al. [17,18] developed a two-dimensional finite element model to study molten pool flow behavior in capillary force and thermocapillary force and proposed a dimensionless number called the normalized average displacement of a liquid particle in a single-laser pulse to predict the surface profile. Furthermore, Wang et al. [19] built a model by combining normalized average displacement (NAD) analytical for thermocapillary flow and capillary regime surface prediction methods to valid capillary force and thermocapillary force regimes. This model fully considers the influence of capillary force and thermocapillary force on surface roughness in the laser polishing process but does not quantify capillary force and thermocapillary force in the laser polishing process. To solve this problem, Zhang et al. [20] established a two-dimensional model to reveal surface morphology evolution and the effect of capillary force and thermocapillary force on molten pool flows. However, these numerical models for laser polishing are based on the melting regime, not an over-melt mechanism. Thus, Li et al. [21] built a two-dimensional axisymmetric numerical model to study the morphology evolution of a Ti6Al4V surface and the flow behavior of a molten pool in an over-melt mechanism of laser polishing. Their results reveal that capillary and thermocapillary forces dominate the molten pool flows and play a major role in smoothing the surface when the surface temperature is lower than the evaporating one. In contrast, recoil pressure drives the molten pool to flow, resulting in uneven surface morphology when the surface temperature exceeds the evaporation temperature. In addition, Xu et al. [22,23] developed a numerical model to investigate the formation mechanism underlying the bulge structure in the center of the laser polishing track by adding surface-active elements (oxygen, sulfur, and phosphorus) into the flow model of the molten pool. Their results reveal that thermocapillary forces affected by surface active elements caused the molten material to flow from the edge of the laser polishing track to the center of the laser polishing track due to a positive surface tension gradient. To solve this problem, some measures were proposed to inhibit the formation of bulges by decreasing the content of active elements and choosing a reasonable overlap percentage of polished tracks.

At present, most numerical simulations are based on the mechanism of continuous wave laser polishing, while there are few studies on the mechanism of pulsed laser and dual-beam laser polishing. Temmler [24] and Nusser et al. [25] studied the effect of dual-beam laser parameters on surface roughness. Their results revealed that the dual-beam laser polishing technology decreased the meso-roughness due to longer melt duration and obtained a smoother surface. Although the experimental study of dual-beam laser polishing has developed, there is no reliable theoretical support. Moreover, research on the numerical simulation of laser polishing with a moving laser heat source is limited, and this numerical simulation can better explain the laser polishing process. Thus, this paper established a two-dimensional numerical model to reveal the dual-beam laser polishing mechanism by coupling heat transfer and laminar flow based on moving laser beams.

In the present study, the effects of different laser energy densities and scanning intervals on surface morphology were studied. To further investigate the mechanism of dual-beam laser polishing, a numerical model was established by coupling heat transfer and laminar flow. The surface morphology evolution and molten pool flow behavior were predicted using the finite element method. Furthermore, the mechanism of dual-beam laser polishing was explained using the temperature field, velocity field, and surface tension. The results revealed that capillary force and thermocapillary force play an important role in the flow of the molten pool. A comparison between the simulation results and the experimental results proved that the numerical model was correct.

## 2. Experiments Study

### 2.1. Experimental Setup and Materials

The experimental equipment used in this study consists of a continuous-wave (CW) fiber laser at the wavelength of 1064 nm (Model: MFSC-1000, Chuangxin Laser Co., Ltd., Shenzhen, China) with a maximum power of 1000 W and a pulse fiber laser at the wavelength of 1064 nm (Model: YDFLP-CL-300-10-W, Hymson Laser Technology Group Co., Ltd., Shenzhen, China) with a maximum power of 300 W, which were used for dual-beam (DB) polishing experiments. Figure 1 shows a schematic diagram and the experimental equipment. The overall system mainly consists of a continuous laser, pulse laser, beam expanding mirror, scanning galvanometer, and a water cooling and sealed process chamber. The maximum speed of the scanning galvanometer (Model: F20PRO-3-MS, Fretak Laser Technology Co., Ltd., Jiangsu, China) with a focal length of 720 mm is 4000 mm/s, and the maximum scanning area is 600 mm × 600 mm. Argon gas with a purity of 99.99% was injected into the sealed processing chamber as a protective gas to avoid oxidation.

The morphology of the Ti6Al4V powders observed using scanning electron microscopy (SEM) is shown in Figure 2a. It is obvious that the powders have good liquidity due to their excellent roundness. A statistical analysis of the diameter of Ti6Al4V powders found that they have a log-normal distribution, as shown in Figure 2b, and the size of powders is distributed mainly at 42 ± 15 μm. In Figure 2b, the abscissa represents powder particle size, the ordinate on the left represents the density distribution of powder particle size, and the ordinate on the left represents the cumulative distribution of powder particle size, which refers to the ratio of the total volume of particles with particle size less than X to the total volume of all particles. The size of the powder particle with a 50% cumulative distribution is 37.5 μm. The Ti6Al4V samples with a size of 100 mm × 100 mm × 8 mm were fabricated using selective laser melting, and the area of single-laser polishing is 10 mm × 10 mm. The surface impurities were scrubbed out with 95% alcohol. Finally, the samples were directly used for the experiment without sandblasting or other treatments.

### 2.2. Experimental Methods

As shown in Figure 3, the laser-polishing experiments contain two steps. First (see Figure 3a), a full factorial design of experiments was carried out to investigate the effects of CW laser energy density (24 J/mm^2^, 28 J/mm^2^, 32 J/mm^2^) and scanning interval (0.1 mm, 0.12 mm, 0.14 mm) on the surface roughness of the selected laser-melted Ti6Al4V samples. During the second step (see Figure 3b), the same full factorial design of experiments was conducted to investigate the effects of pulsed laser energy densities (6 J/cm^2^, 8 J/cm^2^, 10 J/cm^2^) and scanning interval (0.06 mm, 0.08 mm, 0.10 mm) on the best-polished samples obtained in the first stage. The variables investigated in the laser-polishing experiment are listed in Table 2. In addition, the focal spot diameter of the continuous wave laser was about 0.27 mm, exhibiting a top-hat intensity distribution. The focal spot diameter of the pulse laser was about 0.346 mm, exhibiting a Gaussian intensity distribution. The repetition rate and pulse duration of the pulse laser were 100 KHz and 240 ns, respectively.

In this study, a white light interferometer (Model: BRUKER WYKO Contour GT-K, Billerica, MA, USA) was chosen to measure the three-dimensional morphology and surface roughness of Ti6Al4V by scanning an area of 5 mm × 5 mm. A scanning electron microscope (SEM, model: ZEISS Gemini 300, Gina, Germany) was used to observe the morphology of the molten pool. The size of the Ti6Al4V powder particle was also investigated using a laser particle size analyzer (Model: HELOS-0157, SYMPATEC GmbH, Clausthal-Zellerfeld, Germany).

### 2.3. Experimental Results Analysis

Figure 4a shows the influence of laser density and scanning interval on average surface roughness Sa. The results show that with an increase in laser density from 24 J/mm^2^ to 32 J/mm^2^, the surface roughness Sa shows a “V” type variation trend, which is considered as developed re-melting of material. When the CW laser energy density is too great, the molten pool oscillates, which leads to increases in the surface roughness of the Ti6Al4V alloy. When the CW laser energy density is too low, there is insufficient melting of the sample surface, which reduces the polishing effect. In addition, it is noted that at the fixed energy density Ed of 24 J/mm^2^ and 32 J/mm^2^, the influence of the scanning interval from 0.1 mm to 0.14 mm on average roughness Sa is slight. The optimized energy density and scanning interval on average roughness Sa are 28 J/mm^2^ and 0.12 mm, respectively, as obtained from nine tests and shown in Figure 4a. In addition, the average roughness Sa of the sample surface with optimized CW laser parameters reduces by 72.8–83.4% compared with the as-received samples. Figure 4b shows the effects of pulse laser energy density and scanning spacing on the CW laser-polished Ti6Al4V with optimized parameters. The effect plot shows that average roughness increased with the increase in Ed from 6 J/cm^2^ to 10 J/cm^2^, differing from the CW laser-polished sample. The minimum average roughness Sa further decreases to 1.421 μm with a reduction rate of 21.8% compared with the CW laser-polished sample under the pulse laser energy density Ed of 6 J/cm^2^ and scanning intervals of 0.08 mm. Figure 5 shows two-dimensional profiles of the initial, CW laser-polished, and dual-beam (DB) laser-polished surfaces. It is obvious that the surface profiles are smoother when polished with the CW laser and DB laser compared with the initial surface profiles. The spatial spectrum of the original, CW laser-polished, and the DB laser-polished surfaces in the x and y directions are shown in Figure 6. The maximum amplitude of the Fourier component on the surface of Ti6Al4V decreases to 1.6 μm and 1.5 μm, respectively, in the x direction and y direction after dual-beam laser polishing. This suggests that dual-beam laser polishing is useful for improving surface roughness. The three-dimensional morphology of the initial surface and dual-beam laser-polished surface is shown in Figure 7. It can be seen that the peak and valley height of the DB laser-polished Ti6Al4V is greatly decreases, and the average roughness is drops from 10.96 μm to 1.42 μm with a reduction rate of 87%. Accordingly, the maximum height of peak and valley (Rt) also decline from 126.057 μm to 47.299 μm, which shows that surface quality can be improved effectively under the optimized DB laser processing parameters.

## 3. Numerical Model

A two-dimensional numerical model involving heat transfer and laminar flow is developed to study the evolution of surface morphology and the flow behavior of molten pools. To simplify the model and reduce the computational cost, the following assumptions are made for heat transfer and laminar flow [26].

The flow in the melting pool is considered to be incompressible Newton laminar flow.

To simplify the model, the material of the sample is regarded as homogeneous and isotropic.

The laser polishing was carried out below the boiling temperature of Ti6Al4V without considering the evaporation of the model.

Chemical reactions during processing are ignored because laser polishing is carried out in a 99% argon environment.

The absorption rate of laser radiation on the material surface is assumed to be a constant value; however, the angle of laser radiation acting on the material surface changes constantly during the processing.

### 3.1. Properties of the Material

The temperature dependence of dynamic viscosity, specific heat, density, and thermal conductivity was considered, as shown in Figure 8 [17,27,28]. Other property parameters of Ti6Al4V are listed in Table 3 [20,21,22,27,29].

### 3.2. Governing Equations

#### 3.2.1. Heat Transfer

The traditional computational fluid dynamics method was used to solve the problem in this simulation. The heat transfer equation for a material surface governed by Fourier’s law is described as follows [30]:(1)$$\rho C_{eqp}\frac{\partial T}{\partial t}+\rho C_{eqp}\nabla \cdot \left(uT\right)-\nabla \cdot \left(k\nabla T\right)=Q_{\mathrm{source}}$$
where ρ is the density, T is the temperature, t is the time, u is the velocity of molten pool flow, k is the thermal conductivity, $Q_{\mathrm{source}}$ is the laser heat source, and $C_{eqp}$ is the modified heat capacity, which is defined as [31,32]:(2)$$C_{eqp}=C_{p}+L_{m}(\frac{df_{L}}{dT})$$
where $C_{p}$ is the specific capacity of solid materials, $L_{m}$ is the latent heat of melting, and $f_{L}$ is the liquid fraction, which can be described as [32]:(3)$$f_{L}=\{\begin{array}{cc} 0 & T\leq T_{S}\\ \frac{T-T_{S}}{T_{L}-T_{S}} & T_{S}\leq T\leq T_{L}\\ 1 & T\geq T_{L} \end{array}$$
where $T_{S}$ (1877 K) and $T_{L}$ (1900 K) are the solid and liquid temperatures of Ti6Al4V, respectively.

#### 3.2.2. Laminar Flow

The Navier–Stokes equation (N-S equation) for incompressible laminar flow is used to solve the flow velocity of the molten pool by coupling heat transfer and laminar flow. The laminar flow equations on the material surface can be written as:(4)$$\rho \frac{du}{dt}+\rho (u\cdot \nabla )u=\nabla \cdot [-pI+\mu (\nabla u+{\left(\nabla u\right)}^{T})]+F+\rho g$$
(5)ρ∇⋅(u)=0
where p is the pressure, μ is the dynamic viscosity, I is the identity matrix, and $F_{V}$ is the body force, which is defined as:(6)$$F_{V}=F_{g}+F_{b}=\rho_{ref}g(1-\beta (T-T_{ref}))$$
where $\rho_{ref}$ is the reference density, g is the gravity, β is the thermal conductivity, and $T_{ref}$ is the reference temperature.

### 3.3. Boundary Conditions

#### 3.3.1. Heat Transfer

The distance from 0 mm to 1200 mm on the initial two-dimensional surface morphology obtained from the white light interferometer (see Figure 5) was imported into the multi-physics field simulation software COMSOL Multiphysics (Version number: 5.6.0.401, 5 May 2021) as the initial sample surface for simulation. The geometry model with all the boundaries in the simulation is marked, as displayed in Figure 9. Boundary 2 is formed by the heat flux, natural heat convection, and surface heat radiation, which can be expressed as:(7)$$-k\nabla T=\alpha Q+h(T-T_{amb})+\varepsilon \sigma (T^{4}-T_{amb}^{4})$$
where α is the absorption of material, h is the convective coefficient, ε is the radiation emissivity, and σ is the Stefan–Boltzmann constant. In this model, the CW laser with top-hat energy distribution is applied to the material surface, and then the pulse laser with a Gaussian energy distribution is applied to the material surface after cooling for 500 ms. Therefore, the heat flux on boundary 2 is written as [29]:(8)$$f_{\left(x\right)}=\{\begin{array}{cc} 0 & \left|x\right|>x_{0}\\ 1 & \left|x\right|\leq x_{0} \end{array}$$
(9)$$x=x_{1}-vt-x_{0}$$
(10)$$Q_{1}=\frac{P_{1}}{\pi r_{1}{}^{2}}\times f_{\left(x\right)}$$
(11)$$Q_{2}=\frac{2P_{2}}{\pi r_{2}{}^{2}}\exp[\frac{-2{\left(x-x_{0}-V_{pulse}t\right)}^{2}}{r_{2}^{2}}]\times f_{\left(x\right)}$$
where $Q_{1}$ is the CW laser heat source, $P_{1}$ is the CW laser power, $r_{1}$ is the CW laser spot, x is the position of the CW laser, $x_{0}$ is the beginning position of the CW laser, and v and t are the velocity and time of the CW laser, respectively. Similarly, $Q_{2}$ is the pulse laser heat source with a Gaussian distribution, $P_{2}$ is the pulse laser power, $r_{2}$ is the radius of the pulse laser, and $V_{pulse}$ is the velocity of the pulse laser.

Boundaries 1 and 3 include natural heat convection and surface heat radiation, which are expressed as shown in [33]:(12)$$-k\nabla T=h(T-T_{amb})+\varepsilon \sigma (T^{4}-T_{amb}^{4})$$

Boundary 4 is thermal insulation, which is defined as:(13)∇T=0

#### 3.3.2. Laminar Flow

As shown in Figure 9, boundaries 1, 3, and 4 are the non-slip walls, and the top surface is the free deformation surface. The main forces driving the molten pool flow in the free deformation surface mainly include surface tension in the normal direction ($\sigma_{n}$) and the Marangoni force in the tangential direction ($\sigma_{t}$), which is expressed as [34]:(14)$$\sigma =\sigma_{\nu }+\sigma_{\tau }$$
(15)$$\sigma_{n}=k\gamma n=(\nabla \cdot n)\gamma n=(\frac{\partial_{n1}}{\partial x}+\frac{\partial_{n2}}{\partial y})\gamma n$$
(16)$$\sigma_{t}=\nabla \gamma =\frac{\partial \gamma }{\partial T}\nabla T_{t}\cdot t$$
where k is the curvature of the free surface, n is the normal vector, γ is the surface tension coefficient, and $\frac{\partial \gamma }{\partial T}$ is the surface tension gradient, which has an important influence on the molten pool flows. If the surface tension gradient is positive ($\frac{\partial \gamma }{\partial T}>0$), the material will flow from the center of the molten pool with high surface tension to the edge with low surface tension. However, if the surface tension gradient is negative ($\frac{\partial \gamma }{\partial T}<0$), the melt pool flows in the opposite direction [35].

The Marangoni effect is applied to boundary 2 as the tangential force. While the normal force equation is achieved with the weak contribution. The theoretical derivation is expressed as follows.
(17)$$\begin{array}{l} \int_{S}u\sigma_{n}dS=\int_{S}\kappa \gamma u\cdot ndS=\gamma \int_{S}\left(-\nabla \cdot n\right)u\cdot ndS\\ =-\gamma [\int_{S}\left(u\cdot n\right)dI-\int_{S}\left(\nabla u\right)\cdot dS]\\ =\int_{S}\left(\nabla u\right)\cdot dS-\int_{\partial S}\gamma (u\cdot n))dI \end{array}$$
where $\int_{S}\left(\nabla u\right)\cdot dS$ can be written in COMSOL weak form format, and a detailed implementation is covered in ref. [30]. Our integral $\int_{\partial S}\gamma (u\cdot n))dI$ is equal to zero.

The physical boundary conditions set for this numerical simulation are listed in Table 4.

### 3.4. Mesh and Configurations

To improve the simulation accuracy and shorten the simulation time of the molten pool, the influence of the air domain on the simulation results and the resistance of the air to the molten pool flows can be ignored. An arbitrary Lagrange–Eulerian method (ALE) was used to describe the displacement accuracy of the molten pool. The normal velocity of free interface mesh moving equals the normal velocity of molten pool flow, which can be derived as follows [36]:(18)$${\left(x_{t},y_{t}\right)}^{T}\cdot n=u\cdot n$$
where ${\left(x_{t},y_{t}\right)}^{T}$ is the velocity of mesh and u is the velocity of the molten pool.

Free triangular mesh is applied to the whole model, as displayed in Figure 10. In the process of molten pool simulation, the free triangular mesh plays an important role in simulation results for raising the quality of the mesh. To accurately characterize the evolution of the surface morphology of the material and the flow state of the molten pool, boundary 2 is meshed with the more refined element. The grid smoothing algorithm used for the model is the Laplace method. The parameters of the mesh size mentioned above are listed in Table 5. To prevent diffusivity caused by excessive mesh deformation during melt pool flow, automatic mesh redrawing is applied. The grid of the model includes 35,507 triangular elements and 1090 boundary elements. The direct solver used for simulation is PARDISO in the CW laser-polishing process due to its rapidness and small memory occupation, while the indirect solver used in the pulse laser-polishing process is MUMP, due to its greater solving memory for supporting cluster computing. The calculation time took around 45.1 h on a computer with an AMD Ryzen i7-2700 eight-core processor at 3.20 GHz.

## 4. Simulation Results Analysis

In the present study, a CW laser heat source with an energy density Ed of 28 J/mm^2^ was selected to smooth the Ti6Al4V alloy surface. Figure 10 shows the temperature distribution of the material surface under a moving CW laser source. It can be seen that the maximum temperature reached 1933 K at the center of the laser spot when the time reached 0.18 ms. As time increased, the molten pool area where the temperature increased beyond liquidus temperatures extended. Up to 10 ms, the molten pool area reached a maximum because more laser energy accumulated. However, the maximum temperature of the material surface decreased gradually to near the melting temperature, and the molten pool disappeared at t = 20 ms. For cooling 500 ms at room temperature, the maximum temperature was further reduced to 880 K. Meanwhile, a pulse laser beam with an energy density Ed of 6 J/mm^2^ and a scanning velocity of 60 mm/s was chosen to act on the material surface. The temperature distribution of the material surface is shown in Figure 11b. Note that a similar temperature distribution was obtained using the pulse laser polishing process. At t = 525.06 ms, the material surface temperature reached about 2360 K. As time increased, the maximum temperature gradually decreased. Finally, the material surface completely solidified at t = 540 ms because the maximum temperature was lower than the melting temperature (Tm).

As a physical quantity, spatial curvature is used to characterize the top surface of a molten pool, which is expressed as [37]
(19)$$\kappa =-\frac{r(\partial^{2}\phi /\partial r^{2})+\partial \phi /\partial r(1+{\left(\partial \phi /\partial r\right)}^{2})}{r+{\left(1+{\left(\partial \phi /\partial r\right)}^{2}\right)}^{3/2}}$$
where $\frac{\partial \phi }{\partial r}$ is the vertical displacement of the top surface of a molten pool. Figure 12 shows the curvature evolution of the top surface at different times. The results show that the space curvature of the material is significantly reduced after CW laser polishing (t = 520 ms) compared with the initial surface. It can be clearly seen that there was a sudden change in the curvature at r = 100 μm, r = 700 μm, and r = 890 μm. This is because the flow velocity of the molten pool suddenly changes to zero, causing the surface to not be completely smooth at this time. However, the space curvature of the pulse laser polishing-processed surface (t = 540 ms) shows regular periodic fluctuations significantly. The melt, solidify, melt again, and solidify again of material surface in a very short time eventually leads to the formation of spatial curvature cycle fluctuations. Although the new irregularity surfaces are produced using DB laser polishing, the overall surface roughness is lower than the CW laser-polished sample.

The evolution of surface morphology, the temperature field, and the velocity field at different times (t = 5 ms, t = 10 ms, t = 15 ms, t = 20 ms) is illustrated in Figure 13. To describe the flow behavior of the molten pool on the whole model surface, a heating time of 20 ms was selected, at which the morphology of the molten pool gradually became smooth. The fusion boundary is shown as the green contour in Figure 13. The area above the green contour (1900 K) is the molten pool area, and the one below is the solid material. The arrow indicates the flow velocity of the molten pool. The larger the arrow is, the faster the molten pool flow velocity is. It can be found that the vectors are almost invisible in the solid phase, which validates the viscosity model. The molten material flows from the high-temperature center to the low-temperature periphery, as shown by the direction of velocity in Figure 13a. This is because the surface tension in the edge area of the molten pool is higher than the center parts since the surface tension coefficient of Ti6Al4V is negative. Moreover, the material at the bottom of the molten pool flows from the edge to the center due to gravity, and then it flows from the bottom of the molten pool center to the top surface, eventually forming a circulation on the left half of the molten pool, as shown in Figure 13. However, the flow direction of the material changes on the left side of the molten pool. The reason is that the low surface tension caused by a large temperature gradient on the left side of the molten pool cannot overcome gravity, causing the molten pool material to flow from the edge to the center. As the laser beam moves forward, the width and depth of the molten pool on the material surface rapidly increase and then gradually decrease, which matches well with Figure 11. The surface features with large curvature are smoothed continuously with the molten pool flow. Up to 20 ms, the protrusions and pits basically disappear, as shown in Figure 13d. It is pertinent to note that the solidified surface still has a tiny bulge structure because the molten pool has solidified before smoothing as the molten pool moves forward.

In a previous study, it was found that there are two major forces driving the flow of the molten pool: surface tension and the Marangoni effect. If the thermocapillary force introduced by a temperature gradient cannot overcome the viscous force, the capillary drives the flow of the molten pool. Otherwise, the thermocapillary force dominates the molten pool [20].

Figure 14 shows the distribution of the driving force of the molten pool at different times. The black and red curves in Figure 14a–d represent the capillary force and thermocapillary force, respectively. For the surface features with large curvature, the capillary force plays the leading role in reducing the surface roughness. Otherwise, the thermocapillary force related to the temperature gradient smooths the molten pool morphology. It can be found that an obvious smoothness effect appeared on the surface of the molten pool, where 100 μm < r < 400 μm, because the capillary force associated with surface curvature reduced significantly, as shown in Figure 14a. With a temperature increase, the melting area extended, and the dynamic decreased. Up to t = 10 ms, the thermocapillary force came the viscous force of the molten pool to dominate the flow of the molten pool at the edge region where 600 μm < r < 630 μm, which is the thermocapillary force regime. Otherwise, the capillary force was the main driving force, called the capillary force regime, as shown in Figure 14b. It also can be found that the maximum capillary force of the liquid phase occurred at the center of the molten pool due to the large spatial curvature, while the maximum thermocapillary force of the liquid phase occurred at the edge of the molten pool due to the large temperature gradient. Until 20 ms, the effect of the thermocapillary force on the flow behavior of the molten pool reduced due to the lower temperature gradient, and the capillary force was the main force driving the molten pool flows, as displayed in Figure 14d. It was observed that the amplitude of the surface undulation decreased with the evolution of the molten pool. The phenomenon was because the sinusoidal wave was damped by the capillary force and the viscous force of the molten pool [38].

The temperature of the surface decreased to 878 K at t = 520 ms. Meanwhile, a pulse laser beam acted on the top surface of Ti6Al4V. Figure 15 illustrates the surface morphology evolution and molten pool flow behavior at different times. It was obvious that the molten pool dimension formed during pulsed laser polishing was much smaller than that formed during CW laser processing. Moreover, the molten pool flow behavior was similar and flowed from the center to the edge due to the negative surface tension coefficient. At t = 525.37 μm, the depth and width of the molten pool were 9 μm and 255 μm, respectively, while the maximum flow velocity of the molten pool reached 0.64 m/s, as presented in Figure 15a. At this moment, the thermocapillary force dominated the molten pool flows where 250 μm < r < 255 μm and 338 μm < r < 350 μm. Up to 535.77 ms, it was obvious that the liquid phase volume and flow velocity of the molten pool gradually reduced, while the surface temperature increased slightly. In addition, the small liquid phase volume and low temperature led to a lower temperature gradient along the center to the edge of the molten pool, which weakened the Marangoni convection in the tangential. Therefore, the effect of thermocapillary force on the flow of the molten pool could be ignored, and the capillary force dominated the flow of the molten pool, as presented in Figure 16c. The same trend indicating a change in molten pool flow and driving force occurred on the top surface at 539.56 ms. Furthermore, it also can be seen that the capillary force driving the molten pool flow fluctuates significantly. The phenomenon was well matched with the spatial curvature at t = 540 ms (see Figure 12). This demonstrates that the fluctuation in spatial curvature results in a fluctuation in capillary force.

In order to verify the simulation results, laser polishing experiments are carried out using a continuous wave laser with an energy density Ed of 24 J/mm^2^ and a scanning interval of 0.12 mm and a pulse laser with an energy density Ed of 6 J/cm^2^ and a scanning interval of 0.08 mm. The morphology of the molten pool obtained using the experiments and simulations is displayed in Figure 17. The width and depth of the molten pool are 337.59 μm and 63.75 μm, respectively, in the DB laser polishing experiment, as depicted in Figure 17a. It can be seen that the ratio of width to depth of the molten pool is large, which is mainly due to the fact that the laser heat source belongs to the top-hat light with uniform energy distribution. The molten pool is in a completely stable state at t = 5 ms. Figure 17b illustrates the simulated surface profile obtained using the finite element method at t = 5 ms. The red region is the liquid phase, the blue region is the solid phase, and the region between blue and red is the solid–liquid mixing phase. The simulated molten pool width is 404.72 μm, and the error is controlled within 16.6%. The molten pool depth is 51.42 μm, and the error is controlled within 19.3%. This is because the absorption used in the simulation is a fixed value. However, the actual absorption varies with surface morphology, which leads to greater pool depth. This result indicates that the width error of the molten pool has a higher accuracy compared with the depth.

## 5. Conclusions

In this paper, a 2D numerical model was developed to examine the effect of dual-beam laser polishing on the characteristics of a molten pool. In addition, the influence of CW laser polishing and DB laser polishing on the surface roughness was investigated. The conclusions are as follows:The surface roughness of the as-fabricated Ti6Al4V sample is 10.96 μm, which was decreased to 1.818 μm using CW laser polishing processing. Furthermore, the surface roughness of the dual-beam laser-polished Ti6Al4V sample was reduced to 1.421 μm, which was a 21% reduction compared with CW laser polishing.During the CW laser polishing process, the capillary force was the main force that eliminated surface asperities with larger curvature, while the thermocapillary force was the main driving force that smoothed the surface at the edge of the molten pool. However, during the pulse laser polishing process, the effect of thermocapillary force on the molten pool flows was slight due to the small molten pool dimension and lower surface temperature gradient, where the capillary force dominated the molten pool flows.Based on the comparison between the experimental results and the simulated results, the width and depth error of the molten pool was controlled within 16.6% and 19.3%. It was proven that surface morphology evolution and the flow behavior of molten pool were accurate.

## Figures and Tables

**Figure 1 micromachines-14-01765-f001:**
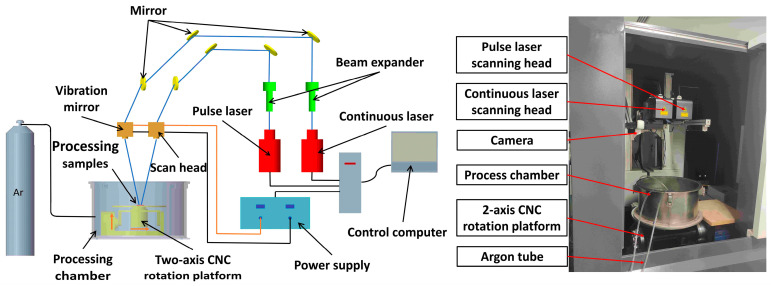
Schematic diagram and experimental setup of laser polishing (LP).

**Figure 2 micromachines-14-01765-f002:**
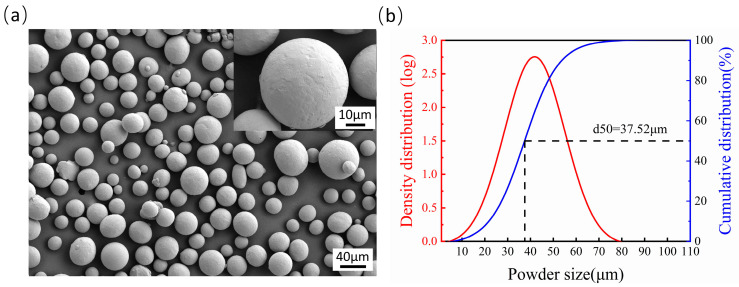
(**a**) SEM and (**b**) size distribution of the Ti6Al4V powder.

**Figure 3 micromachines-14-01765-f003:**
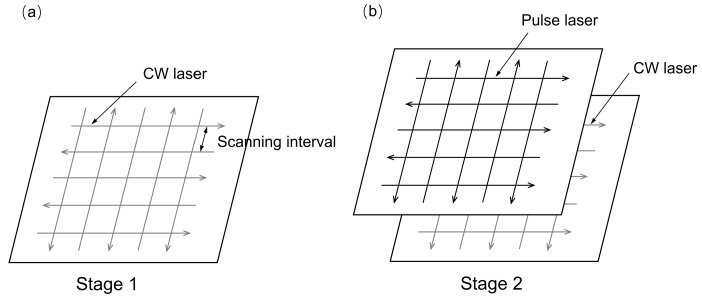
Schematic illustration showing laser polishing. (**a**) The CW laser scan strategy and (**b**) the dual-beam (DB) laser scan strategy.

**Figure 4 micromachines-14-01765-f004:**
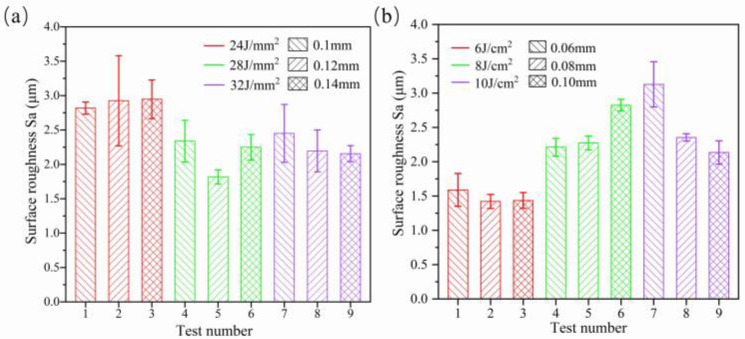
Average Ra calculated from the recorded surface roughness data at three locations of the surfaces (**a**) polished with different CW laser energy influences and scanning intervals and (**b**) polished with different pulse energy influences and scanning intervals in E_CW_ = 28 J/mm^2^ and D_CW_ = 0.12 mm.

**Figure 5 micromachines-14-01765-f005:**
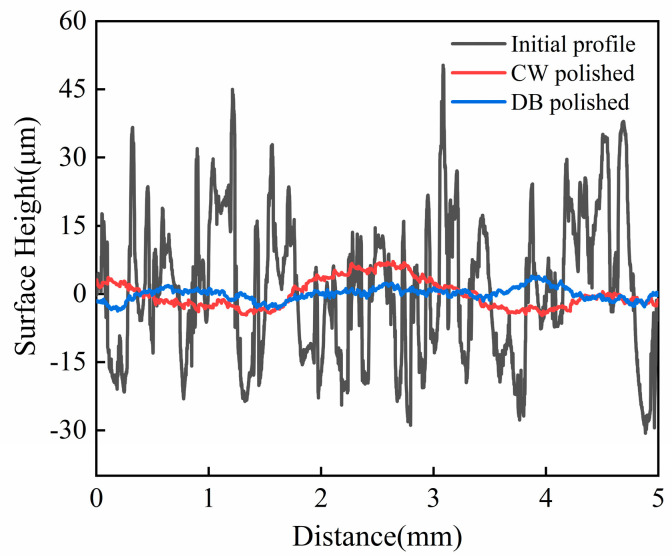
Comparison of the initial profile, CW laser-polished profile, and DB laser-polished profile of additive manufacturing Ti6Al4V.

**Figure 6 micromachines-14-01765-f006:**
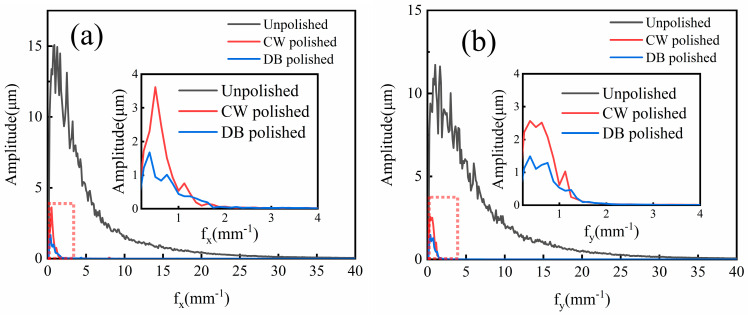
Overlapped two-dimensional frequency spectra of unpolished, CW laser-polished, and dual-beam (DB) laser-polished surfaces in the (**a**) x-direction and (**b**) y-direction.

**Figure 7 micromachines-14-01765-f007:**
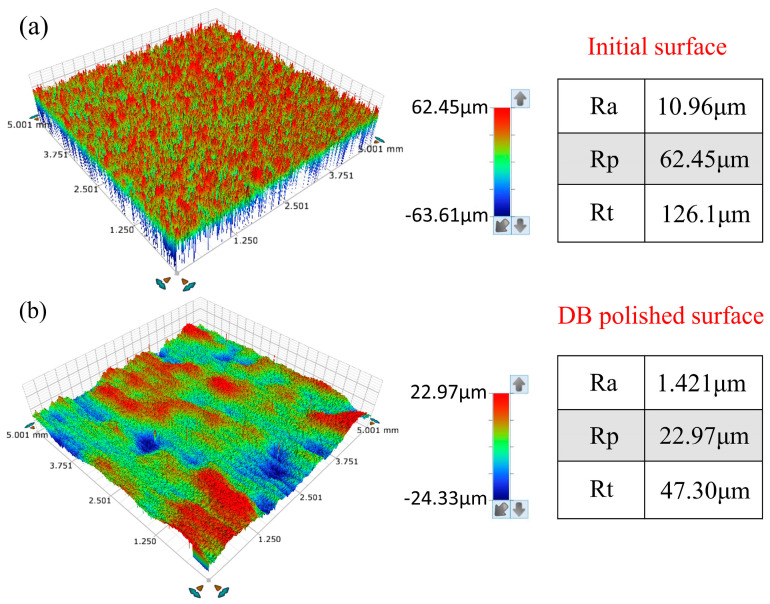
Stereoscopic surface morphology of the (**a**) initial and (**b**) dual-beam (DB) laser-polished surfaces.

**Figure 8 micromachines-14-01765-f008:**
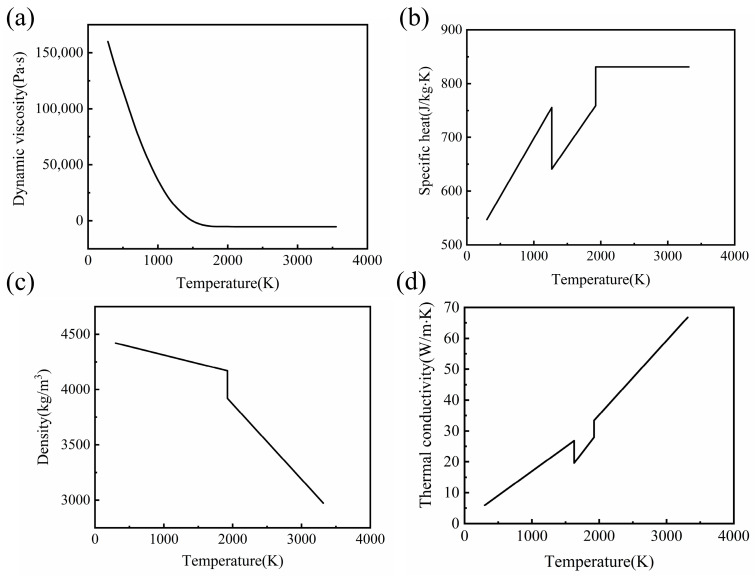
Thermal physical parameters of additive manufacturing Ti6Al4V. (**a**) Dynamic viscosity, (**b**) specific heat, (**c**) density, and (**d**) thermal conductivity.

**Figure 9 micromachines-14-01765-f009:**
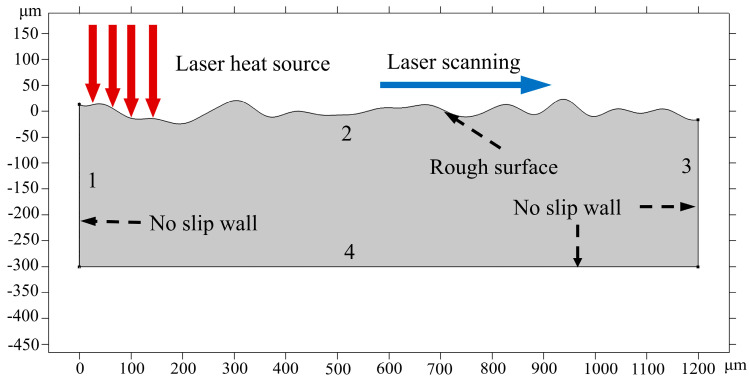
Geometric model.

**Figure 10 micromachines-14-01765-f010:**
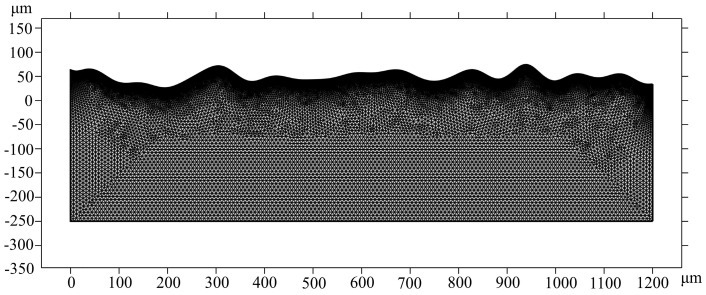
Meshed geometry.

**Figure 11 micromachines-14-01765-f011:**
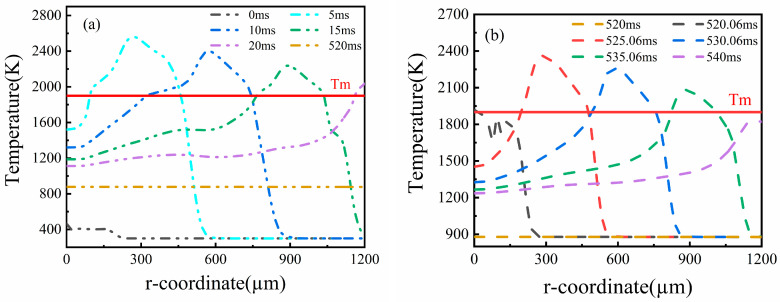
Temperature distribution of the Ti6Al4V surface at different times during (**a**) continuous wave (CW) laser polishing and (**b**) dual-beam (DB) laser polishing.

**Figure 12 micromachines-14-01765-f012:**
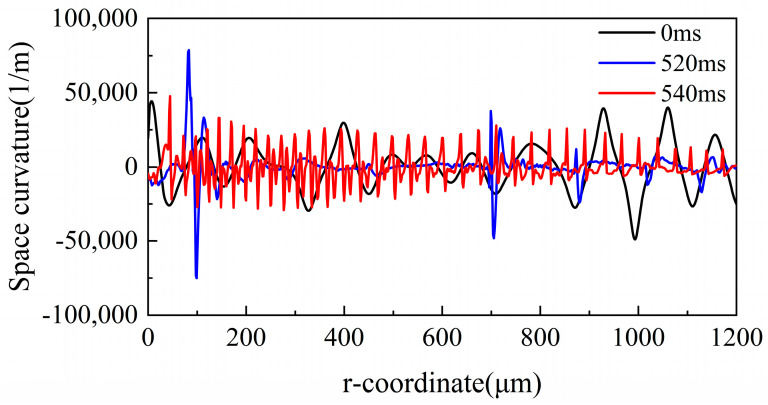
Space curvature evolution of the Ti6Al4V surface at t = 0 ms, t = 520 ms, and t = 540 ms.

**Figure 13 micromachines-14-01765-f013:**
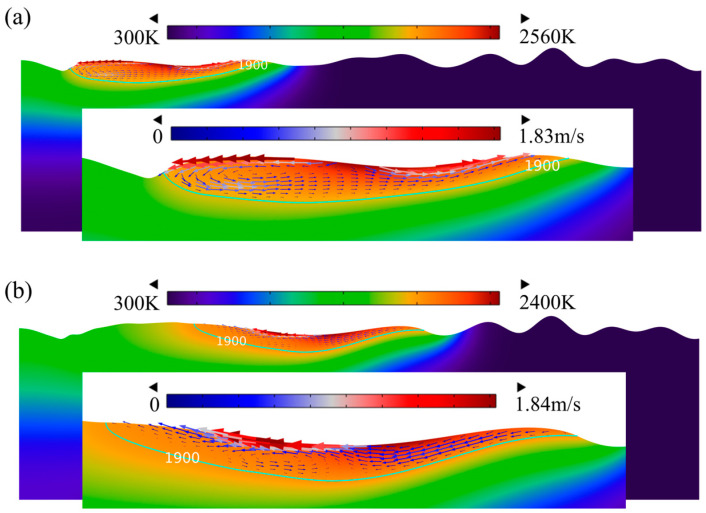
Surface morphology evolution during CW laser polishing at (**a**) t = 5 ms, (**b**) t = 10 ms, (**c**) t = 15 ms, and (**d**) t = 20 ms.

**Figure 14 micromachines-14-01765-f014:**
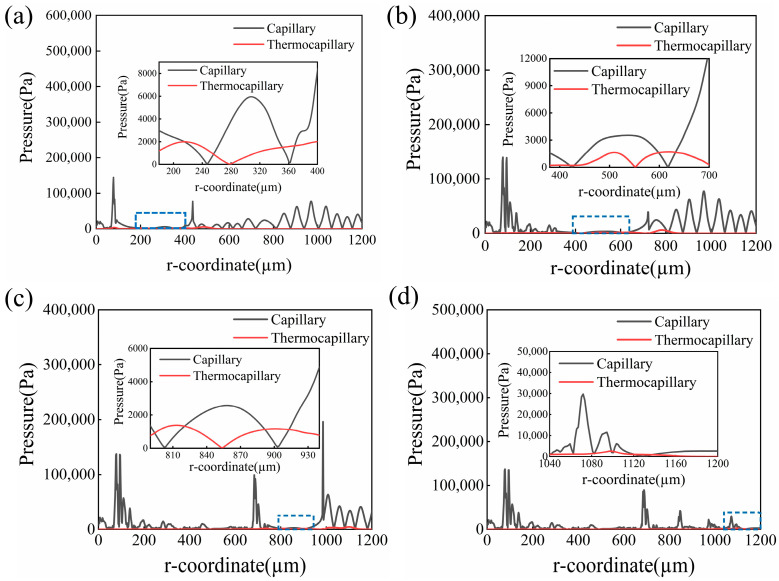
Variation in the capillary force and the thermal capillary force at different times during CW laser polishing at (**a**) t = 5 ms, (**b**) t = 10 ms, (**c**) t = 15 ms, and (**d**) t = 20 ms.

**Figure 15 micromachines-14-01765-f015:**
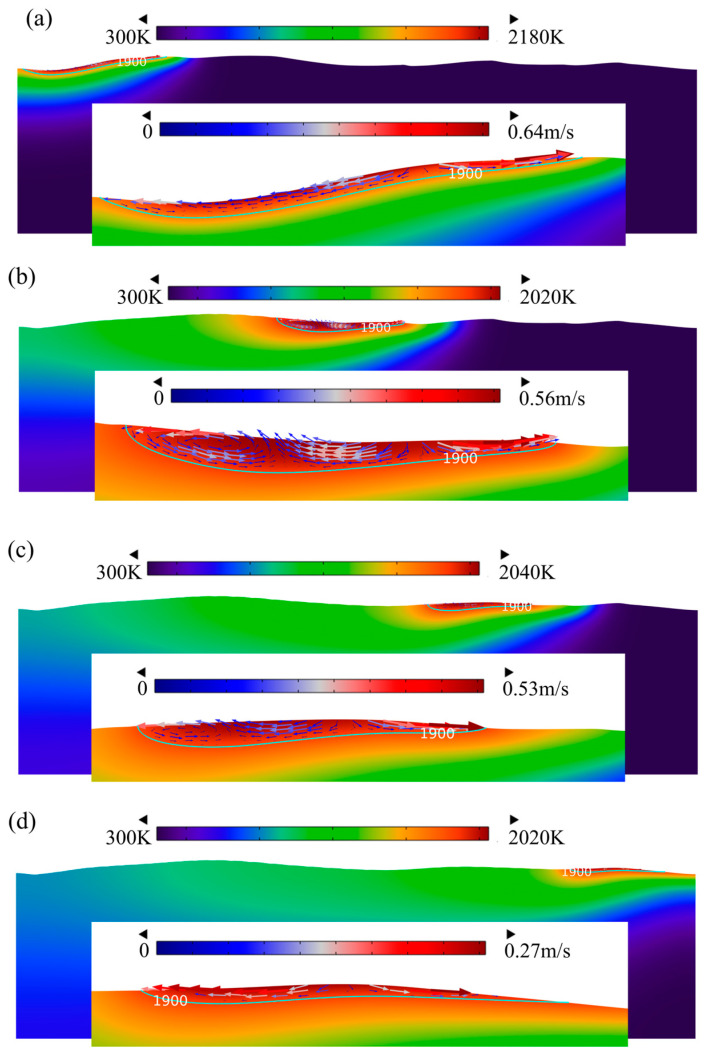
Surface morphology evolution during pulse laser polishing at (**a**) t = 525.37 ms, (**b**) t = 530.98 ms, (**c**) t = 535.77 ms, and (**d**) t = 539.56 ms.

**Figure 16 micromachines-14-01765-f016:**
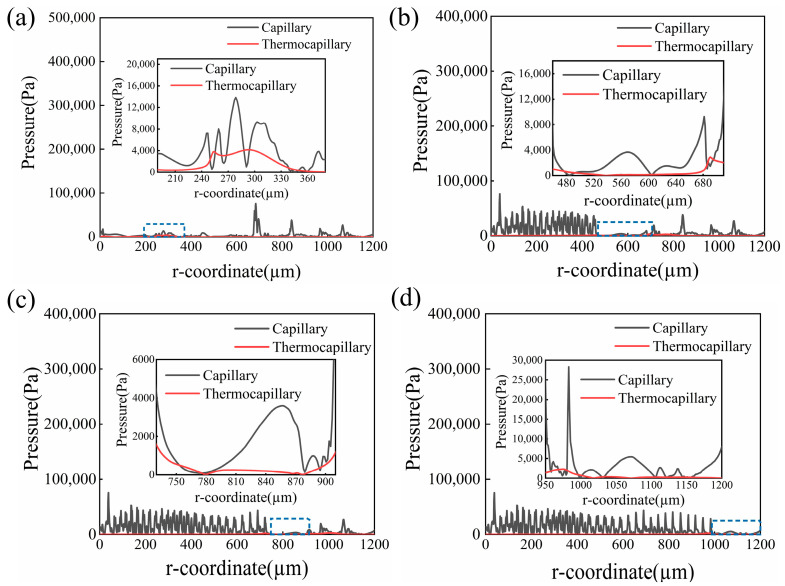
Variation in the capillary force and the thermocapillary force at different times during pulse laser polishing at (**a**) t = 525.37 ms, (**b**) t = 530.98 ms, (**c**) t = 535.77 ms, and (**d**) t = 539.56 ms.

**Figure 17 micromachines-14-01765-f017:**
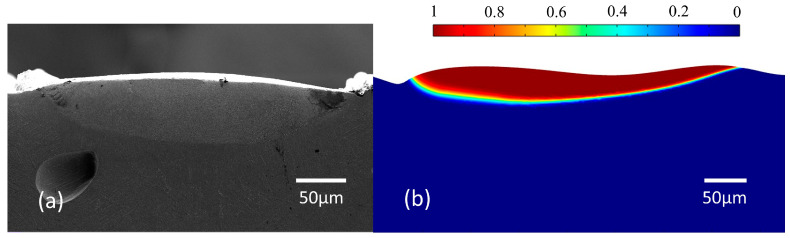
Comparison between the experimental and simulated surface profiles. (**a**) Experimental surface profile and (**b**) simulated surface morphology.

**Table 1 micromachines-14-01765-t001:** Comparison among laser polishing and other finishing process methods.

Surface Machining Method	Material Applicability	Adaptability to Complicated External Surface	Surface Quality	Reference
Laser polishing	Wide	High	6.62 μm to 0.55 μm, and improving the microhardness	[11,12]
Multi-jet polishing	Wide	High	0.84 μm to 0.03 μm	[13]
Electrochemical polishing	Narrow	High	17 μm to 0.25 μm	[14]
Bonnet polishing	Medium	Medium	104 nm to 3.7 μm	[15]
Conventional surface machining (grinding/milling)	Wide	Low	<10 nm (Sa), but complex working procedure	[16]

**Table 2 micromachines-14-01765-t002:** Variables investigated as LP process parameters.

Stage 1			
Processing parameters			
CW laser energy density (J/mm^2^)	24	28	32
CW laser scanning interval (mm)	0.1	0.12	0.14
Stage 2			
Processing parameters			
CW laser energy density (J/mm^2^)	24	28	32
CW laser scanning interval (mm)	0.1	0.12	0.14
Pulse laser energy density (J/cm^2^)	6	8	10
Pulse laser scanning interval (mm)	0.06	0.08	0.1

**Table 3 micromachines-14-01765-t003:** Fixed physical property parameters of additive manufacturing Ti6Al4V.

Parameter	Nomenclature	Value
Solidus temperature (K)	$T_{s}$	1877
Liquids temperature (K)	$T_{l}$	1923
Melting temperature (K)	$T_{m}$	1900
Boiling temperature (K)	$T_{v}$	3315
Thermal expansion coefficient	β	$1.1\times 10^{5}$
Absorptivity	α	0.3
Latent heat of melting (J/kg)	$L_{m}$	$2.86\times 10^{5}$
Convective coefficient (W/(m∙K))	h	10
Temperature derivative of surface	∂γ/∂T	$-2.8\times 10^{-4}$

**Table 4 micromachines-14-01765-t004:** The main physical boundary conditions.

Physical Condition	Boundary	Boundary Condition
Laser radiation	2	Heat flux
Heat convection	1, 2, 3	Convection
Radiation	1, 2, 3	Radiation
Normal stress	2	Weak contribution
Tangential stress	2	Marangoni effects
No slip wall	1, 3, 4	Wall

**Table 5 micromachines-14-01765-t005:** Parameters of triangular mesh.

Parameter (Unit)	Top Surface	The Rest
Maximum mesh size (μm)	1.6	8
Minimum mesh size (μm)	0.008	0.2
Maximum mesh growth rate	1.05	1.1
Curvature factor	0.2	0.2

## Data Availability

The data presented in this study are available on request from the corresponding author.

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
