# Peer review of "Numerical and Experimental Analysis of Dual-Beam Laser Polishing Additive Manufacturing Ti6Al4V"

_micromachines, 2023, doi:10.3390/mi14091765_

Round 1

Reviewer 1 Report

1. The introduction needs to be revised carefully. Please refer to some recent high-quality journal papers.

2. The conclusion needs further refinement. The conclusions need to reflect some of the most critical conclusions.

3. Please provide a detailed description of laser parameters, such as laser beam spot diameter, frequency, and other data.

4. The size and clarity of images in Figure 1, Figure 2, Figure 5, Figure 6, Figure 7, Figure 8, Figure 9, Figure 10, Figure 11, Figure 12, Figure 13, Figure 14, Figure 15, and Figure 16 are suggested to be further improved. All pixels of these images need to be very clear when displayed at a 100% scale.

5. “The roughness of surface is decreases initially before increasing with the raise from 24 J/mm2 to 32 J/mm2 of CW laser energy density.” “J/mm2 to 32 J/mm2” should be 24 J/mm2 to 32 J/mm2. Please check the whole manuscript for such issues and make corrections and improvements.

6. The experimental parameters corresponding to Figure 17 need to be clearly listed.

7. Please check and improve the English expressions all through the manuscript.

8. It is recommended to discuss the theoretical and technical significance of this work in the conclusion of this article.

 Please check and improve the English expressions all through the manuscript.

Author Response

Dear Reviewer,

Thank you for giving us the opportunity to submit a revised draft of the manuscript “Numerical and experimental analysis of dual-beam laser polishing additive manufacturing Ti6Al4V” for publication in the Journal of Micromachines. We appreciate the time and effort that you dedicated to providing feedback on our manuscript and are grateful for the insightful comments on and valuable improvements to our paper.

We have incorporated most of the suggestions made by the reviewers. Those changes are highlighted within the manuscript. Please see below, in black, for a point-by-point response to the reviewers’comments and concerns. All page numbers refer to the refer to the revised manuscript file with tracked changes.

1 The introduction needs to be revised carefully. Please refer to some recent high-quality journal papers.

Responds: Thank you for this suggestion. I have revised the introduction carefully and added some recent high-quality journal papers. 

2 The conclusion needs further refinement. The conclusions need to reflect some of the most critical conclusions.

Responds: Thank you for this suggestion. I have revised it in the paper.

3 Please provide a detailed description of laser parameters, such as laser beam spot diameter, frequency, and other data.

Responds: Thank you for this suggestion. I have revised it in the paper. (line157)

4 The size and clarity of images in Figure 1, Figure 2, Figure 5, Figure 6, Figure 7, Figure 8, Figure 9, Figure 10, Figure 11, Figure 12, Figure 13, Figure 14, Figure 15, and Figure 16 are suggested to be further improved. All pixels of these images need to be very clear when displayed at a 100% scale.

Responds: I am really sorry for the quality of those pictures were poor when displayed at a 100% scale. I have provided clearer pictures in the paper.

5 “The roughness of surface is decreases initially before increasing with the raise from 24 J/mm2 to 32 J/mm2 of CW laser energy density.” “J/mm2 to 32 J/mm2” should be “24 J/mm2 to 32 J/mm2”. Please check the whole manuscript for such issues and make corrections and improvements.

Responds: I am really sorry for such issues. I have revised it in the paper. (line510)

6 The experimental parameters corresponding to Figure 17 need to be clearly listed.

Responds: Thank you for this suggestion. I have added the experimental parameters corresponding to Figure 17 in the paper. (line479)

7 Please check and improve the English expressions all through the manuscript.

Responds: Thank you for this suggestion. I have modified the English expression carefully in the paper.

8 It is recommended to discuss the theoretical and technical significance of this work in the conclusion of this article.

Responds: Thank you for this suggestion. I have added the theoretical and technical significance of this work in the conclusion of this article. (line504)

Reviewer 2 Report

A numerical model of dual-beam laser polishing AMed Ti6Al4V was established in this paper, which is helpful to further understand the laser polishign process. However, the innovation point of this process and model is not clearly addressed. 

I would like the authors to address the following concerns before publishing.

1.The literature review should be further improved. There are some other different polishing methods have been developed for the polishing of additively manufactured surfaces, the comparison of laser polishing with them should be briefly introduced in the literature review part, such as multi-jet polishing (Post processing of additively manufactured 316L stainless steel by multi-jet polishing method. Journal of Materials Research and Technology. 2023 Mar 1;23:530-50.), bonnet polishing (Theoretical and experimental investigation of material removal in semi-rigid bonnet polishing of binderless tungsten carbide. Journal of Materials Research and Technology. 2023 May 1;24:1597-611.), magnetic field assisted polishing, abrasive flow machining, electro-chemical finsihing etc.

2. The quality of some figures should be further enahnced, especially the fontsize inside which are difficult to read, such as figs.8,9, 10,14,16

3. Regarding Fig. 17, colorbar should be added to show the meaning of the colour.

Nil

Author Response

Dear Reviewer,

Thank you for giving us the opportunity to submit a revised draft of the manuscript “Numerical and experimental analysis of dual-beam laser polishing additive manufacturing Ti6Al4V” for publication in the Journal of Micromachines. We appreciate the time and effort that you dedicated to providing feedback on our manuscript and are grateful for the insightful comments on and valuable improvements to our paper.

We have incorporated most of the suggestions made by the reviewers. Those changes are highlighted within the manuscript. Please see below, in black, for a point-by-point response to the reviewers’comments and concerns. All page numbers refer to the refer to the revised manuscript file with tracked changes.

1. The literature review should be further improved. There are some other different polishing methods have been developed for the polishing of additively manufactured surfaces, the comparison of laser polishing with them should be briefly introduced in the literature review part, such as multi-jet polishing (Post processing of additively manufactured 316L stainless steel by multi-jet polishing method. Journal of Materials Research and Technology. 2023 Mar 1;23:530-50.), bonnet polishing (Theoretical and experimental investigation of material removal in semi-rigid bonnet polishing of binderless tungsten carbide. Journal of Materials Research and Technology. 2023 May 1;24:1597-611.), magnetic field assisted polishing, abrasive flow machining, electro-chemical finsihing etc.

Responds: Thank you for this suggestion. I have further improved the literature review and added the comparison of the multi-jet polishing, electrochemical polishing, bonnet polishing, the traditional grinding polishing and laser polishing. The results were summarized in Table 1. (line 113)

2. The quality of some figures should be further enahnced, especially the fontsize inside which are difficult to read, such as figs.8,9, 10,14,16

Responds: I am really sorry for the quality of some pictures were poor. I have revised it in the paper.

3. Regarding Fig. 17, colorbar should be added to show the meaning of the colour.

Responds: Thank you for this suggestion. I have added a colorbar in Fig. 17. (line500)

Round 2

Reviewer 1 Report

This research work is valuable.

Minor editing of English language required

Reviewer 2 Report

The paper is acceptable after this round's revision. No more comments. 

Found one typo mistake in the second paragraph of the introduction part.

The word 'virous' should be 'various'. Pls check the paper carefully.